# Aromatic ring-opening metathesis

Valeriia Hutskalova[1] & Christof Sparr[1✉]

Aromatic compounds are used across chemistry and materials science as a result of their stability, characteristic interactions, defined molecular shape and the numerous approaches for their synthesis by a diversity of cyclization reactions[1]. By contrast, the cleavage of inert aromatic carbon–carbon bonds remained largely unfeasible owing to the unfavourable energetics of disrupting aromaticity on ring opening. For non-aromatic structures, alkene metathesis catalysed by transition-metal alkylidenes is established as one of the most versatile carbon–carbon bond-forming and -breaking reactions[2,3]. However, despite remarkable advancements, strategies to open aromatic compounds by metathesis remain elusive[4]. Here we report aromatic ring-opening metathesis (ArROM) for the cleavage of aromatic rings, including tetraphene, naphthalene, indole, benzofuran and phenanthrenes, by using Schrock–Hoveyda molybdenum catalysts. The reactions for each ring system proceed through unique alkylidene intermediates. We further show the possibility for stereoselective ArROM with exquisite catalyst control over the configuration of atropisomers. ArROM is, therefore, a viable and efficient approach to catalytically transform and interconvert various aromatics without the requirement for any reagents or photoexcitation.

Whereas aromatic compounds participate in a multitude of substitution reactions in which the aromaticity of the ring structure is restored[1,5], transformations that permanently rupture aromatic moieties, known as dearomatizations, are recognized as highly challenging[6–11]. Important dearomatization methods comprise the Birch reduction[12], arene hydrogenations[13–16], a diversity of cycloadditions[17,18] and oxidations[19] (such as the formation of quinones). Another type of dearomatization relies on the ring expansion of aromatic substrates, which, however, remained underdeveloped owing to the high dissociation energy of aromatic C=C bonds[20–26]. A common approach for the preparation of seven-membered rings was pioneered by Buchner and is initiated by the (6+1) addition of carbenes[22], nitrenes[23,24] or phosphinidenes[25] followed by a 6π disrotatory electrocyclic ring opening (Fig. 1a). A related methodology was recently developed for cycloadditions with an arenophile followed by epoxidation and ring expansion[26] (Fig. 1b). Meanwhile, methods for cleaving aromatic rings, in which ring opening permits to break the cyclic structure of arenes and heteroarenes, are noticeably scarce. Nature addresses this challenge by subjecting (hetero)arenes to enzymatic oxidations[27]. For instance, bacterial dioxygenases convert arenes to *cis*-dihydrodiols, which are then transformed into muconic acid derivatives on enzymatic cleavage[28] (Fig. 1c), whereas eukaryotes typically use monooxygenase to form arene oxides as the first step of oxidative ring opening[29]. One of the few methods for the non-enzymatic cleavage of aromatic rings is represented by a recently reported copper-catalysed conversion of diversely functionalized arenes (such as anilines and arylboronic acids) into alkenylnitriles through carbon–carbon bond cleavage using $O_2$ and $NaN_3$ as reagents[30,31] (Fig. 1d). Considering the limited range of methods for aromatic ring-opening reactions, there is, hence, a profound need for synthetic approaches to break aromatic structures, ideally taking place without reagents and by a catalytic manifold to regulate reactivity and selectivity.

Owing to its exceptional aptitude and wide range of applications, alkene metathesis catalysed by transition metal alkylidenes gained great importance in organic synthesis[32,33] (Fig. 1e). The cleavage of carbon–carbon multiple bonds by a catalytic process under mild conditions along with the formation of unproblematic side products continues to inspire the broad implementation of this transformation[32]. However, despite the striking advancements of alkene and alkyne metathesis[2,3,33–36], the opening of arenes by metathesis remained unprecedented. Density functional theory calculations of metathesis reactions even suggested the inability of benzene to participate in ring opening owing to the loss of aromaticity in the formation of Ru metallacycles for which highly unfavourable energetics were predicted[4]. Nevertheless, given the importance of metathesis catalysed by transition metal alkylidenes, we questioned if an efficient catalytic system could kinetically and thermodynamically address aromatics for ring-opening metathesis if empowered by a suitable endogenous driving force (Fig. 1f). In particular, we envisaged that the formation of stabilized aromatic ring systems and strain release would enable arene and heteroarene cleavage.

### Aromatic ring-opening metathesis

To assess the viability of ArROM, the [2+2] cycloaddition was, thus, coupled to an arene-forming ring-closing metathesis (RCM) step to promote ring cleavage through a common metallacycle intermediate (Fig. 2a). In a first exploratory study, the reactivity of the readily accessible tetraphene **1** was thereby evaluated in the presence of the commercial molybdenum catalyst **C1**. Here, when using 10 mol% catalyst at 65 °C, the desired product **2** was obtained with a yield of 82%. The thermodynamically favourable formation of product **2** associates with its higher overall aromatic stabilization compared with substrate **1** (for density functional theory calculations, see

[1]Department of Chemistry, University of Basel, Basel, Switzerland. ✉e-mail: christof.sparr@unibas.ch

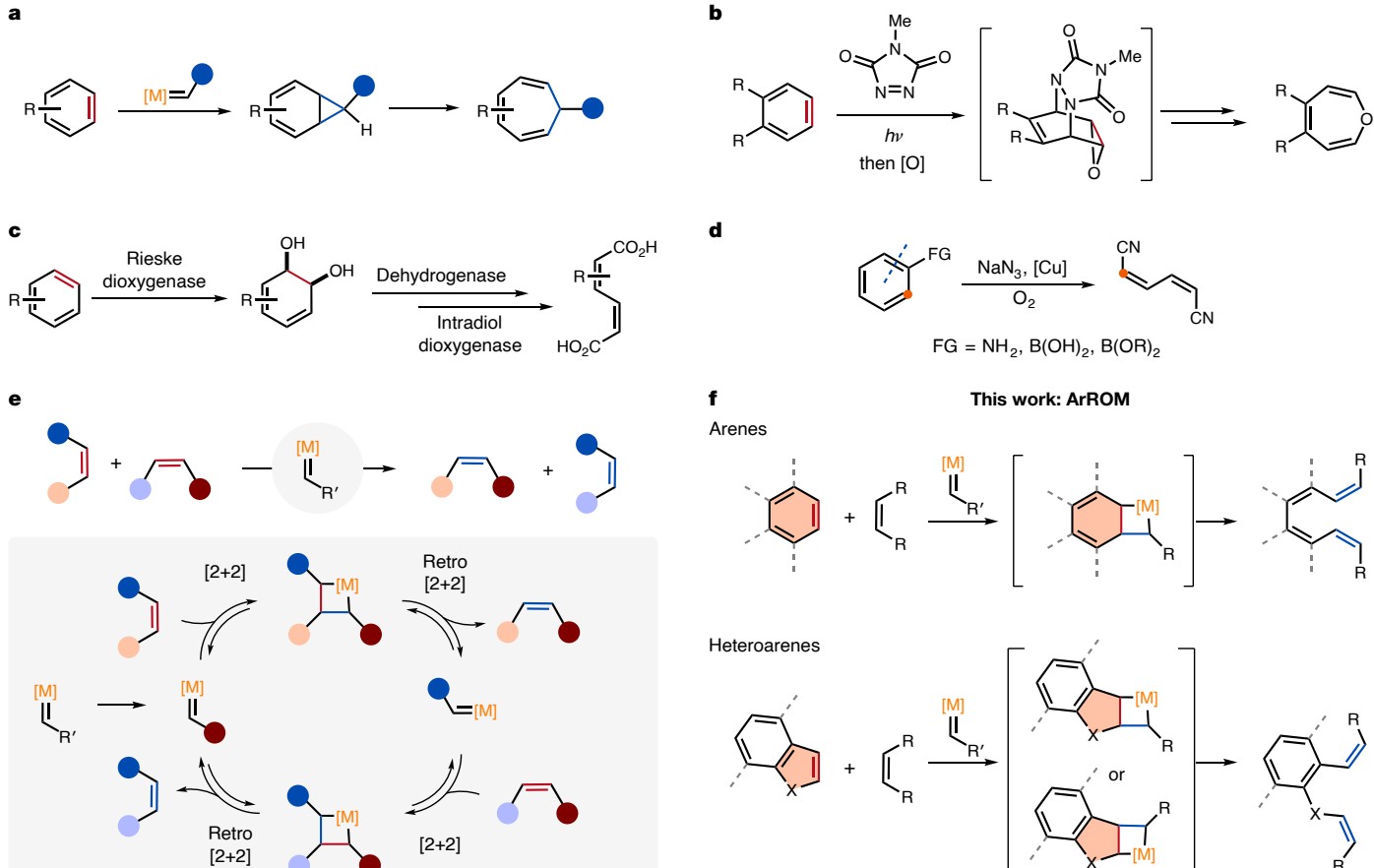

**Fig. 1 | Background and concept. a**, Buchner aromatic ring expansion. **b**, Dearomatization by ring expansion to form oxepines. **c**, Enzymatic cleavage of aromatic rings: arene oxidation pathway in prokaryotes. **d**, Arene ring opening with NaN₃ and O₂ for the synthesis of alkenylnitriles. **e**, General scheme

and mechanism of alkene–alkene metathesis. **f**, Aromatic ring-opening metathesis (ArROM, this work). The red bonds are breaking and the blue bonds are forming. [M], metal; FG, functional group.

Supplementary Table 16). This result served as the first evidence that aromatic rings are addressable by metathesis, constituting our starting point for further exploration of the ring opening of a diversity of aromatics. We next set out to evaluate whether ArROM could break aromatic rings with increased stabilization by drawing our attention to the cleavage of naphthalene substrate **3** (Fig. 2b). For this purpose, ArROM was combined with twofold RCM in a cascade reaction. The obtained alkylidene intermediate was, therefore, expected to further react with the appended alkene to form a terminal aromatic ring, which was observed in the desired chrysene **4** obtained with a yield of 74%, representing an unconventional approach to forge fused polyaromatics. Encouraged by these results, we envisaged the possibility of ArROM for five-membered ring heterocycles (Fig. 2c). To test the reactivity of nitrogen and oxygen heterocycles in heteroaromatic ring-opening metathesis, we first examined the indole substrate **5a**, which was indeed efficiently converted by ArROM with twofold RCM[37] by forming an additional arene as well as a heterocycle. Mild catalytic conditions at 65 °C were sufficient for converting **5a** into the desired naphtho-indole **6a** with a remarkable yield of 98%, in which proton nuclear magnetic resonance (NMR) analysis of the crude reaction mixture confirmed the absence of detectable side products formed in this transformation. Similarly, substrate **5b** bearing a benzofuran ring also effectively provided the phenanthro-furan product **6b** under the same reaction conditions with a yield of 92%. With the possibility of expeditiously creating planar polyaromatics, we explored whether bidirectional synthesis by twofold ArROM and twofold RCM enables the direct generation of extended polycyclic aromatic hydrocarbons (Fig. 2d). Indeed, substrates **7a**

and **7b** efficiently provided arylated benzo[*k*]tetraphenes (**8a**, 67%; **8b**, 56%), whereas the regioisomeric **7c** gave benzo[*m*]tetraphene **8c** with another characteristic geometry (63%). Furthermore, even the highly unstable dibenzo[*a,j*]tetracene **8d** with an expanded polyaromatic core was accessible by twofold ArROM with double RCM. Although the reaction conditions for ArROM emerged as compatible for the synthesis of such a labile compound, the yield was diminished by the spontaneous decomposition of the product, as observed on isolation. Aiming to explore the general potential of ArROM for distinct heteroarene ring-opening transformations, we tested a class of indole substrates for an alternative opening mode of the five-membered rings (Fig. 2e). To assess the feasibility of this opposite indole cleavage reaction, we subjected substrates **9a**–**9i** to ArROM. Because another *N-o*-styryl indole is formed in this transformation, the metathesis reaction is reversible, providing the possibility to equilibrate indoles by ArROM–RCM. The experiments, thus, yielded an equilibrium mixture of the substituted indoles **9a**–**9i** and the corresponding constitutional isomers **10a**–**10i**, enabling the investigation of their relative stabilities. The fluorine substituent in **9d** and **10d** was consequently found to have a minor impact, leading to a 1:1 mixture of these interconverting indoles. However, the introduction of substituents with strong stabilizing and destabilizing effects gave markedly different isomeric ratios. A discernible trend was observed in the ArROM–RCM indole equilibration, revealing that electron-donating groups of indoles **9a** and **9b** shift the ratios towards their corresponding isomers (**10a** and **10b**, respectively). By contrast, the introduction of electron-withdrawing groups (**9f**–**9i**) led to the opposite behaviour, whereas the impact of the substitution

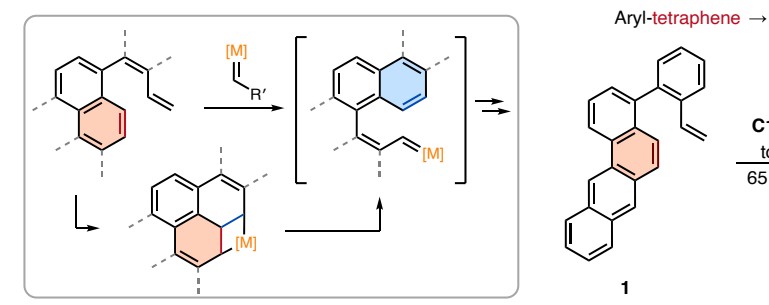
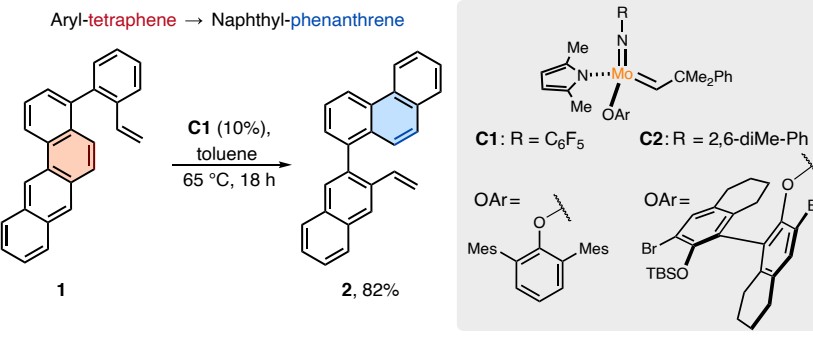

**a** ArROM–RCM cascade

Aryl-tetraphene → Naphthyl-phenanthrene

**C1**: R = C₆F₅  **C2**: R = 2,6-diMe-Ph

**1** → **C1** (10%), toluene, 65 °C, 18 h → **2**, 82%

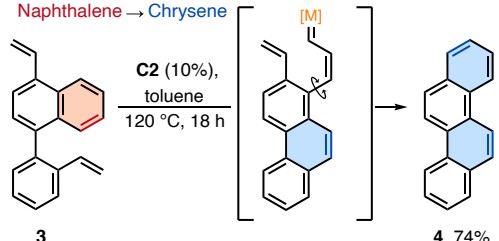
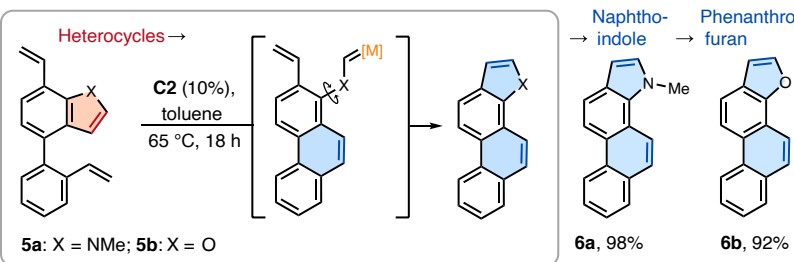

**b** ArROM with twofold RCM for fused aromatics

Naphthalene → Chrysene

**3** → **C2** (10%), toluene, 120 °C, 18 h → **4**, 74%

**c** ArROM with twofold RCM for benzofused five-membered heterocycles

Heterocycles →

**5a**: X = NMe; **5b**: X = O → **C2** (10%), toluene, 65 °C, 18 h

Naphtho- → indole: **6a**, 98%

Phenanthro- → furan: **6b**, 92%

**d** Bidirectional twofold ArROM–twofold RCM to polycyclic aromatic hydrocarbons

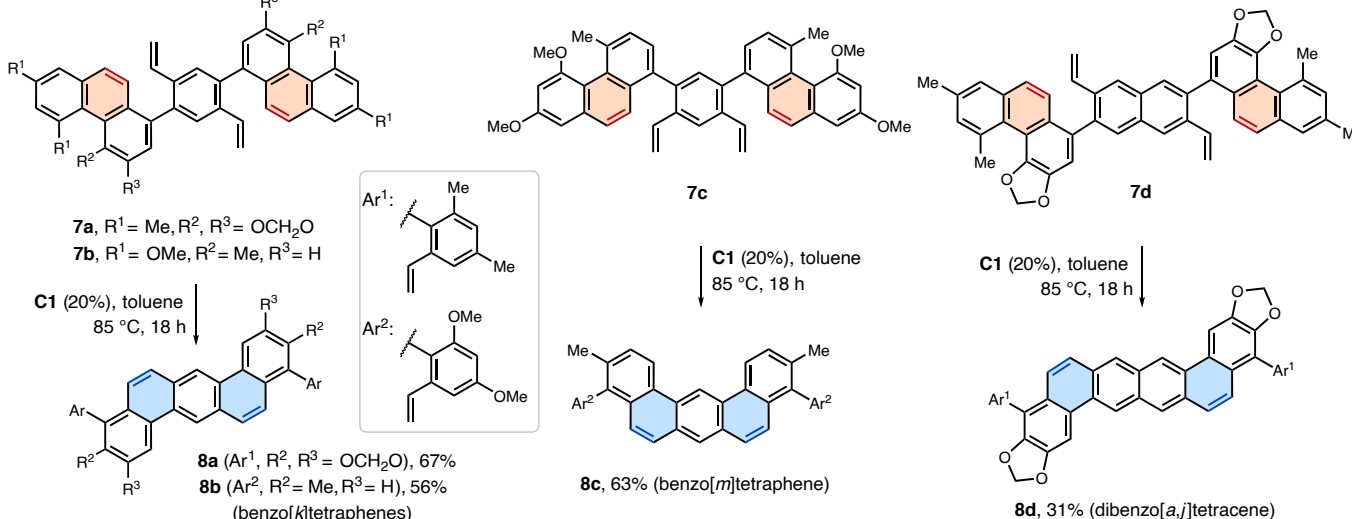

**7a**, R¹ = Me, R² , R³ = OCH₂O
**7b**, R¹ = OMe, R² = Me, R³ = H

Ar¹, Ar²

**7c**

**7d**

**C1** (20%), toluene, 85 °C, 18 h

**8a** (Ar¹, R², R³ = OCH₂O), 67%
**8b** (Ar², R² = Me, R³ = H), 56%
(benzo[k]tetraphenes)

**C1** (20%), toluene, 85 °C, 18 h → **8c**, 63% (benzo[m]tetraphene)

**C1** (20%), toluene, 85 °C, 18 h → **8d**, 31% (dibenzo[a,j]tetracene) Unstable product

**e** Equilibration of indoles by ArROM–RCM

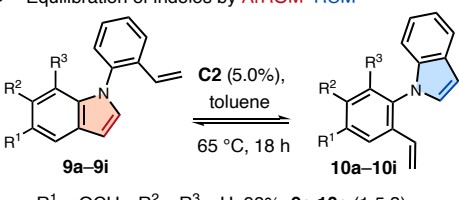

**9a–9i** ⇌ **C2** (5.0%), toluene, 65 °C, 18 h ⇌ **10a–10i**

R¹ = OCH₃, R² = R³ = H: 98%, **9a**:**10a** (1:5.3)
R¹ = CH₃, R² = R³ = H: 97%, **9b**:**10b** (1:2.4)
R¹ = R² = H, R³ = F: 87%, **9c**:**10c** (1:2.2)
R¹ = F, R² = R³ = H: 90%, **9d**:**10d** (1:1)
R¹ = R³ = H, R² = F: 93%, **9e**:**10e** (1:0.5)
R¹ = Cl, R² = R³ = H: 96%, **9f**:**10f** (1:0.4)
R¹ = R³ = H, R² = Cl: 89%, **9g**:**10g** (1:0.3)
R¹ = CF₃, R² = R³ = H: only **9h** detected
R¹ = CO₂Me, R² = R³ = H: only **9i** detected

**f** Twofold ArROM–threefold RCM cascade

Bis(o-styryl)-biindole → Diindolyl-phenanthrene

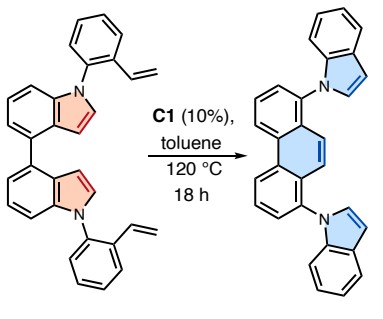

**11** → **C1** (10%), toluene, 120 °C, 18 h → **12**, 41%

**g** ArROM–twofold RCM for N-aryl indoles

**13a–13i** → **C2** (10%), toluene, 65 °C, 18 h → **14a–14i**

**Fig. 2 | Aromatic ring-opening metathesis. a**, Tetraphene. **b**, Naphthalene. **c**, Five-membered heterocycles. **d**, Synthesis of polycyclic aromatic hydrocarbons. **e**, Equilibration of indoles. **f**, Twofold ArROM of indoles (for the intermediates, see Extended Data Fig. 1a). **g**, ArROM for N-aryl indoles (for the scope, see Extended Data Fig. 1b). The red text indicates ring opening and the blue text indicates ring closing. Mes, mesityl; TBS, SiMe₂tBu.

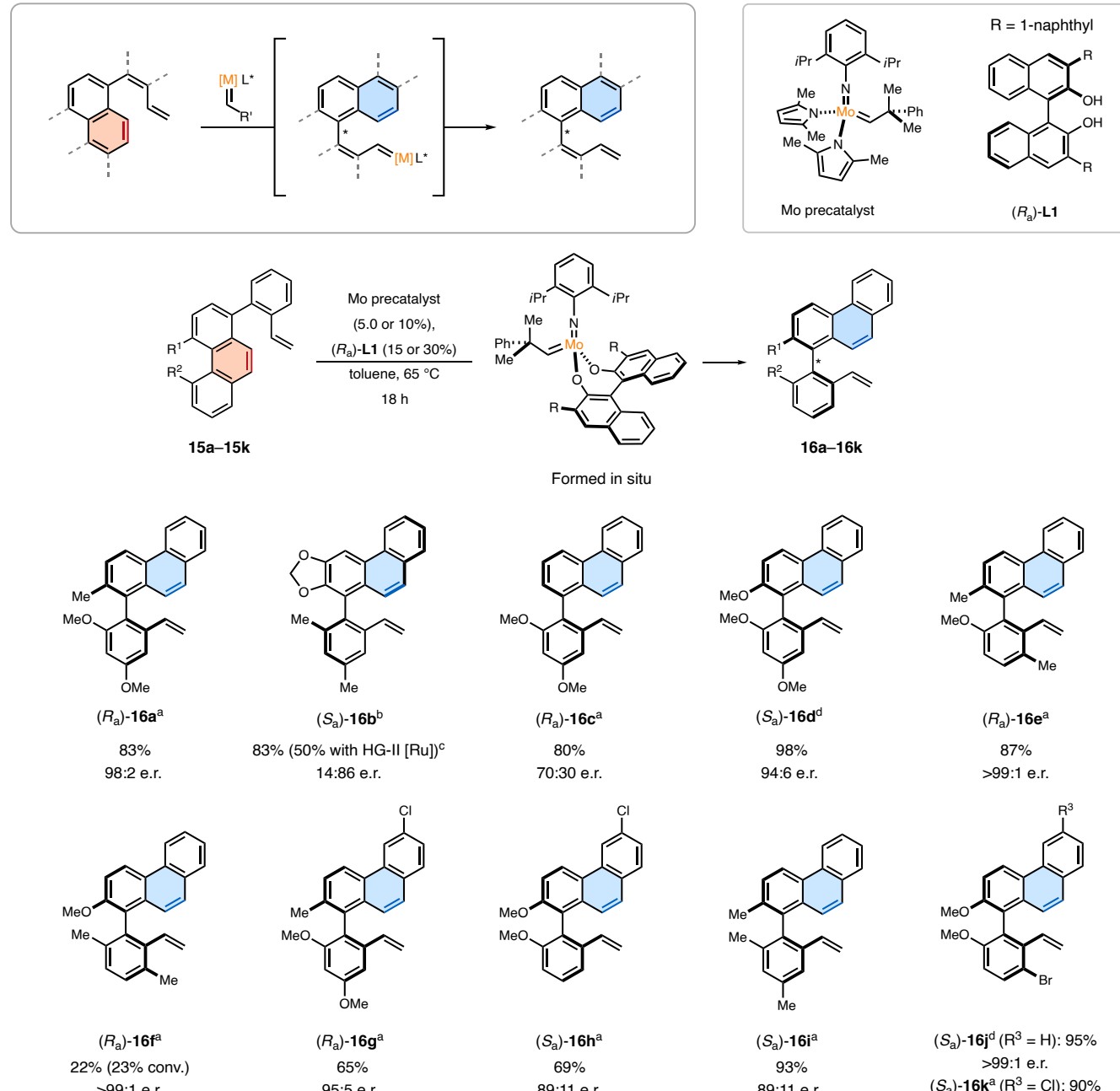

**Fig. 3 | Atroposelective ArROM.** [a]Conditions: phenanthrene substrate **15a–15k** (70.0 μmol), Mo precatalyst (3.50 μmol, 5.0 mol%), ($R_a$)-**L1** (10.5 μmol, 15 mol%), toluene (2.3 ml), 65 °C, 18 h. Yields are given for isolated products; e.r. values were determined of the isolated products using high-performance liquid chromatography on a chiral stationary phase. [b]Reaction performed using an ($S_a$)-configured ligand with R = mesityl. [c]With Hoveyda–Grubbs II (HG-II) catalyst to (±)-**16b**. NMR yield. [d]Mo precatalyst (7.00 μmol, 10 mol%), ($R_a$)-**L1** (21.0 μmol, 30 mol%). L, ligand.

pattern was evident in the equilibration of **9c**–**9e**. This possibility to isomerize indole heterocycles, hence, represents a mechanistically distinct approach to interconvert the constitutional isomers of aromatics. Subsequently, we questioned if complex cascade reactions involving multiple ArROM and RCM steps are also feasible (Fig. 2f and Extended Data Fig. 1a). Specifically, bis-(*o*-styryl)-biindole **11** was expected to transform into the product **12** conceivably through indole ring-opening metathesis to produce an alkylidene intermediate that then participates in the ArROM of the second indole moiety, forming the phenanthrene system. The last step of the cascade comprises a third RCM step leading to the formation of a second indole ring system. Here, with catalyst **C1** at 120 °C, the desired diindolyl-phenanthrene **12** was obtained with 41% yield after an 18 h reaction. This intricate domino reaction underlines

the distinct potential of ArROM in complex transformations by merging several consecutive steps in a catalytic reaction cascade. To evaluate the spectrum of reactions enabled by (hetero)ArROM even more, we studied a third class of indole ring-opening reactions by the combination with twofold RCM (Fig. 2g, Extended Data Fig. 1b and Extended Data Table 1). At the outset of our studies, it was observed that the reaction of **13a** also proceeds with the ruthenium-based Hoveyda–Grubbs II (HG-II) catalyst, giving 47% product, whereas remarkable yields (96%–98%) were achieved with molybdenum catalyst **C2**. ArROM provides the desired phenanthrenyl-indole **14a** after a particularly simple purification procedure and is amenable to 2.0 mol% **C2** on a 112-mg scale (97%). Computational studies aligned with the observed reactivity differences between the Ru and Mo catalysts, with markedly lower activation

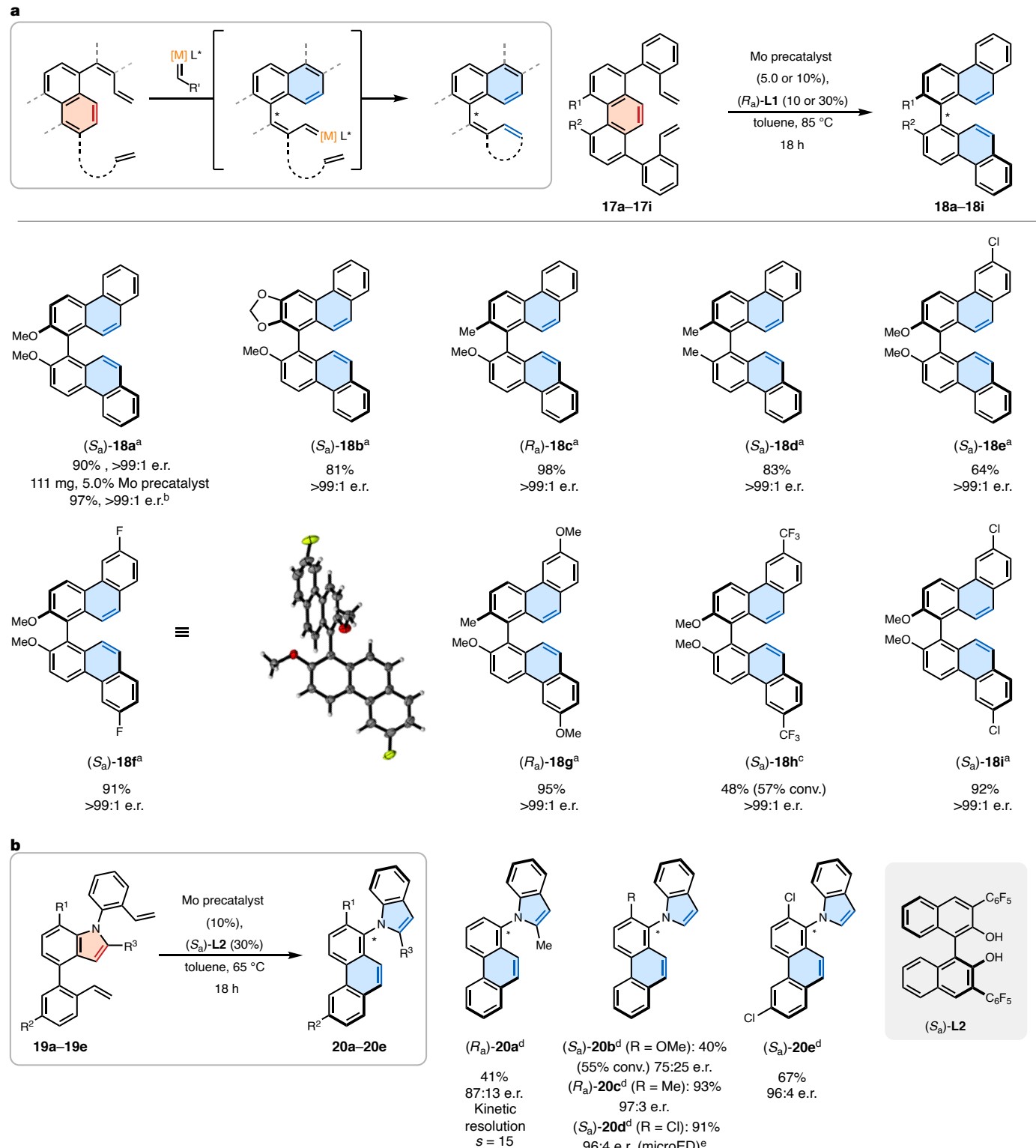

**Fig. 4 | Atroposelective ArROM. a**, AArROM of phenanthrene derivatives with twofold RCM. [a]Conditions: phenanthrene substrate **17a**–**17i** (70.0 μmol), Mo precatalyst (7.00 μmol, 10 mol%), ($R_a$)-**L1** (21.0 μmol, 30 mol%), toluene (2.3 ml), 85 °C, 18 h. [b]Performed using 250 μmol substrate **17a**, Mo precatalyst (12.5 μmol, 5.0 mol%) and ($R_a$)-**L1** (25.0 μmol, 10 mol%). [c]Performed on a 55.0-μmol scale. **b**, AArROM of indoles. [d]Conditions: indole substrate **19a**–**19e**

(70.0 μmol), Mo precatalyst (7.00 μmol, 10 mol%), ($S_a$)-**L2** (21.0 μmol, 30 mol%), toluene (2.3 ml), 65 °C, 18 h. Yields are given for isolated products; e.r. values were determined of the isolated products using high-performance liquid chromatography on a chiral stationary phase. [e]Absolute configuration determined by microED.

energies predicted for the aromatic ring opening to Mo metallacycles (Extended Data Figs. 2 and 3). Notably, the methodology proved to be highly efficient for substrates bearing both electron-donating (**14c**)

and electron-withdrawing (**14b** and **14d**–**14f**) groups. Even a sterically hindered system and a substrate containing an azaindole core engaged in the transformation to form **14g** and **14h**, although with a lower yield

at partial conversion, whereas [4]helicenyl-indole **14i** was forged in a 98% isolated yield.

## Atroposelective ArROM

Inspired by the methods for stereoselective alkene metathesis[38–41] in the context of our recent work on atroposelective arene-forming alkene metathesis[42], we envisaged that arenes could participate in stereoselective ArROM to govern the configuration of emerging stereogenic axes. Owing to the importance of atropisomers, the development of conceptually unique and reliable methods for their stereoselective synthesis is of high interest. To examine the feasibility of atroposelective ArROM (AArROM), we evaluated whether the opening of phenanthrene rings would take place to specifically generate atropisomers with a particularly challenging tetra-*ortho*-substitution (Fig. 3). The choice of a chiral catalyst would, thus, enable control over the configuration of the stereogenic axes by AArROM. By ligand variation (Extended Data Fig. 4 and Extended Data Table 2), chiral catalysts formed in situ using a Mo-dipyrrolyl precursor[43] were assessed for AArROM with substrates **15a–15k**, which are readily accessible by cross-coupling chemistry. We were pleased to find an enantioselectivity of 98:2 for the formation of ($R_a$)-**16a** obtained in 83% yield using ligand ($R_a$)-**L1**. When exploring the generality of the methodology, diverse phenanthrene substrates were successfully subjected to AArROM, affording the corresponding products with high atroposelectivities and often outstanding yields (up to 98%, 50% (±)-**16b** with Hoveyda–Grubbs II [Ru]). Interestingly, a decrease in selectivity was observed for the atropisomer ($R_a$)-**16c** devoid of an otherwise cumbersome fourth biaryl *ortho*-substituent. By contrast, the method was efficient for substrate **15i**, which lacks any potential coordinating substituents, whereas generally high enantioselectivities of up to >99:1 enantiomeric ratio (e.r.) were obtained for the atropisomeric products (($R_a$)-**16e**, ($R_a$)-**16f**, ($R_a$)-**16j** and ($R_a$)-**16k**).

Encouraged by these results, we anticipated to combine AArROM with twofold RCM by transforming substrates **17a–17i** into biphenanthrenes (Fig. 4a). Surprisingly, we discovered that all the tested binaphthol ligands led to an extraordinary performance, yielding the desired products with unexpectedly high enantioselectivities. The substrate scope confirmed the generality of the methodology, revealing that varying the substitution patterns did not have an impact on the selectivities and that all the products were obtained with an enantioenrichment higher than 99:1. Furthermore, the method proved to remain highly efficient with 5.0 mol% precatalyst (10 mol% ($R_a$)-**L1**) on a 111-mg scale (97%, >99:1 e.r. for ($S_a$)-**18a**) or for the substrate **17d** lacking coordinating groups, whereas a single crystal of product ($S_a$)-**18f** enabled the determination of the absolute configuration by X-ray crystallography. On the basis of these findings, together with the feasibility of ArROM of five-membered rings and indoles in particular[44], we envisioned that AArROM could also be developed for heteroaromatic systems (Fig. 4b). To examine this hypothesis, the indole substrate **19a** was first tested to evaluate the possibility of a kinetic resolution by ArROM. Interestingly, when the racemic substrate **19a** with a configurationally stable stereogenic axis was subjected to the reaction with ligand ($S_a$)-**L2**, the desired product **20a** was obtained with an e.r. of 87:13, whereas the starting material **19a** remained with an enantioenrichment of 18:82 ($s$ = 15). By contrast, the stereodynamic substrates **19c** and **19d** led to notably high enantioselectivities (97:3 e.r. and 96:4 e.r.) and yields (93% and 91%) by means of a dynamic kinetic resolution, with the absolute configuration established by microcrystal electron diffraction (microED) analysis of compound ($S_a$)-**20d**.

## Conclusion

We present the viability of ArROM, which encompasses a wide range of reaction manifolds, distinct cascade reactions, equilibrations of heterocycles, kinetic resolutions, bidirectional synthesis of polycyclic aromatic hydrocarbons, dynamic kinetic resolutions and highly atroposelective transformations controlled by chiral Schrock–Hoveyda molybdenum alkylidene catalysts. The strategies were successfully applied to the ring opening of various aromatic systems, including tetraphene, naphthalene, phenanthrenes and five-membered heterocycles of indole and benzofuran substrates. It is anticipated that arene and heteroarene metathesis will markedly widen the capabilities of transition metal alkylidene-catalysed metathesis. The scope of heterocyclic systems anticipated to engage in ArROM renders this strategy particularly appealing for applications in medicinal chemistry, whereas the cleavage of carbocyclic arene rings opens new avenues to create a diversity of complex polyaromatics.

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

## Data availability

The data supporting the findings of this study are available in the article and its Supplementary Information. Supplementary crystallographic data for this paper can be obtained from the Cambridge Crystallographic Data Centre at www.ccdc.cam.ac.uk/structures (CCDC 2358712 and CCDC 2358713).

**Acknowledgements** This project has received funding from the European Research Council (ERC) under the European Union's Horizon 2020 research and innovation programme (grant agreement number 101002471), the Swiss National Science Foundation (10001653) and the Swiss Nanoscience Institute (microED). We thank XiMo Inc. for molybdenum precursors, the Baudoin group for access to gloveboxes, the Häussinger group for NMR support, A. Prescimone for X-ray and microED crystallography, Eldico Scientific for microED measurements, and M. Devereux for computational support.

**Author contributions** C.S. and V.H. conceived the study, designed the experiments and analysed the data. V.H. performed the experiments. C.S. and V.H. wrote the paper.

**Funding** Open access funding provided by University of Basel.

**Competing interests** The authors declare no competing interests.

**Additional information**
**Correspondence and requests for materials** should be addressed to Christof Sparr.

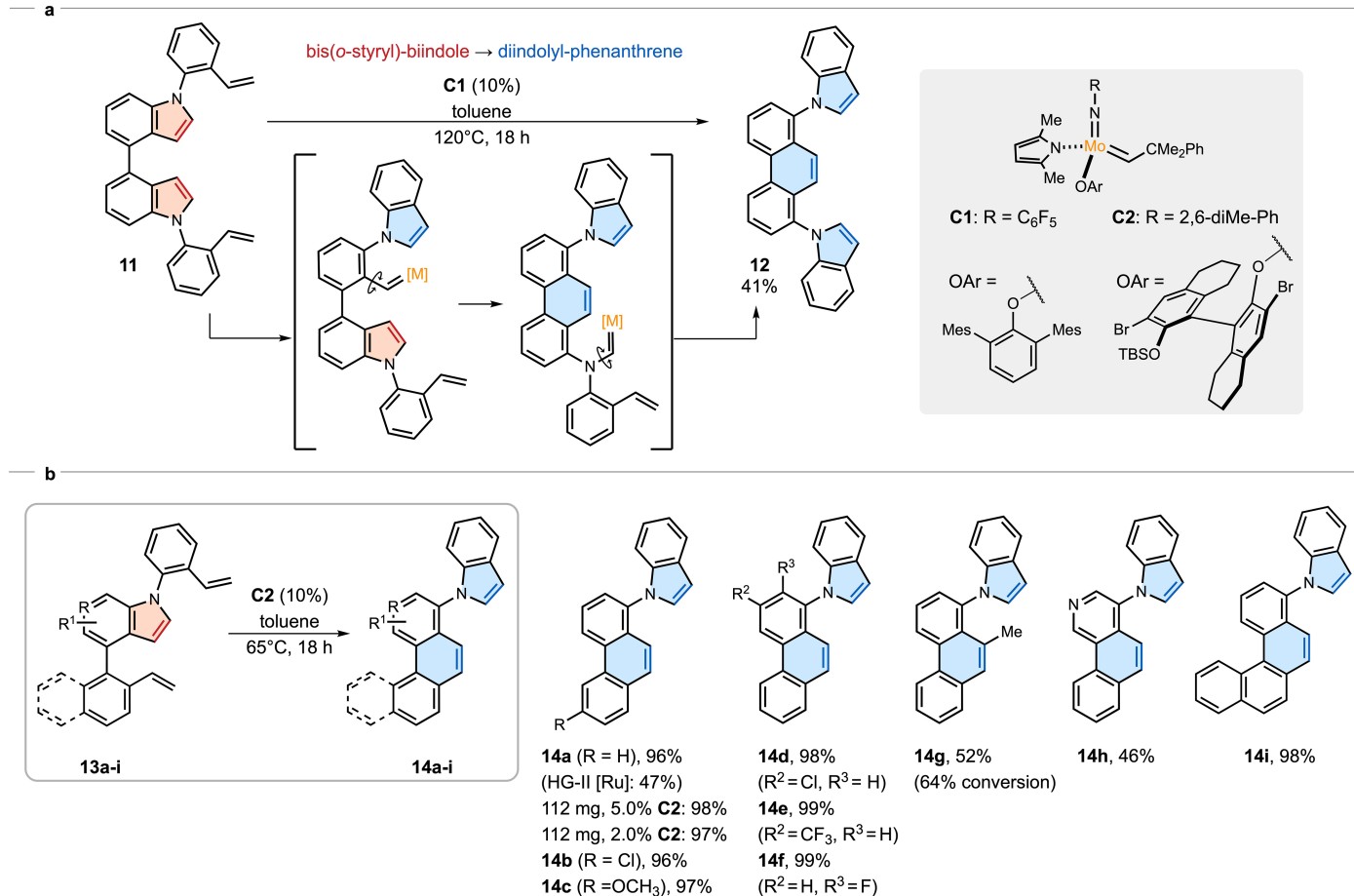

**Extended Data Fig. 1 | ArROM of indoles. a**, Cascade metathesis involving twofold ArROM and threefold RCM. **b**, Substrate scope of ArROM for *N*-aryl indoles.

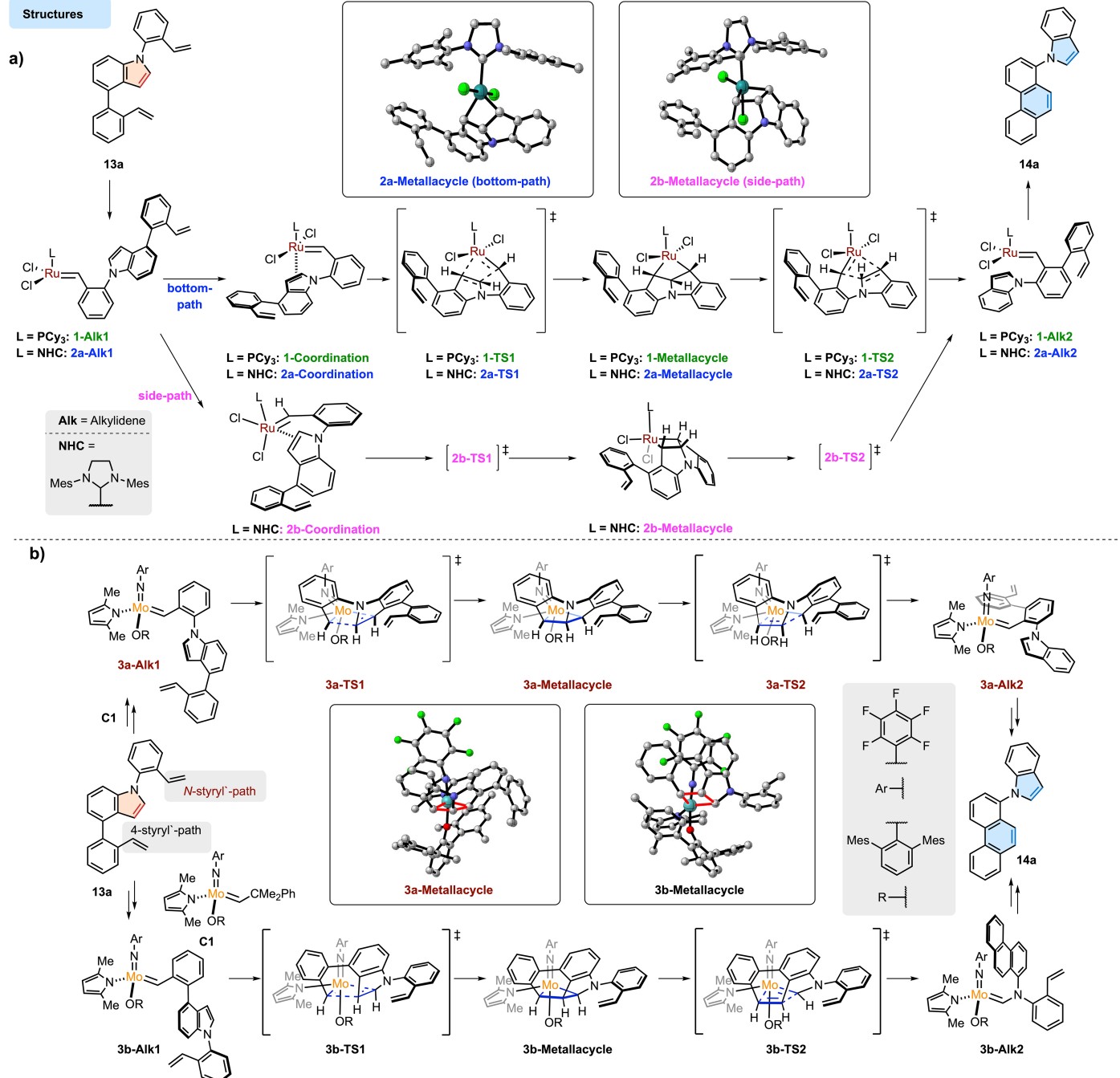

**Extended Data Fig. 2 | Computational studies for ArROM-twofold RCM of substrate 13a for the comparison of different catalytic systems. a**, Ru-catalysed metathesis (Grubbs I and Hoveyda-Grubbs II catalysts): bottom- and side-path mechanisms. **b**, Mo-catalysed metathesis (catalyst **C1**): *N*-styryl'- and 4-styryl'-paths.

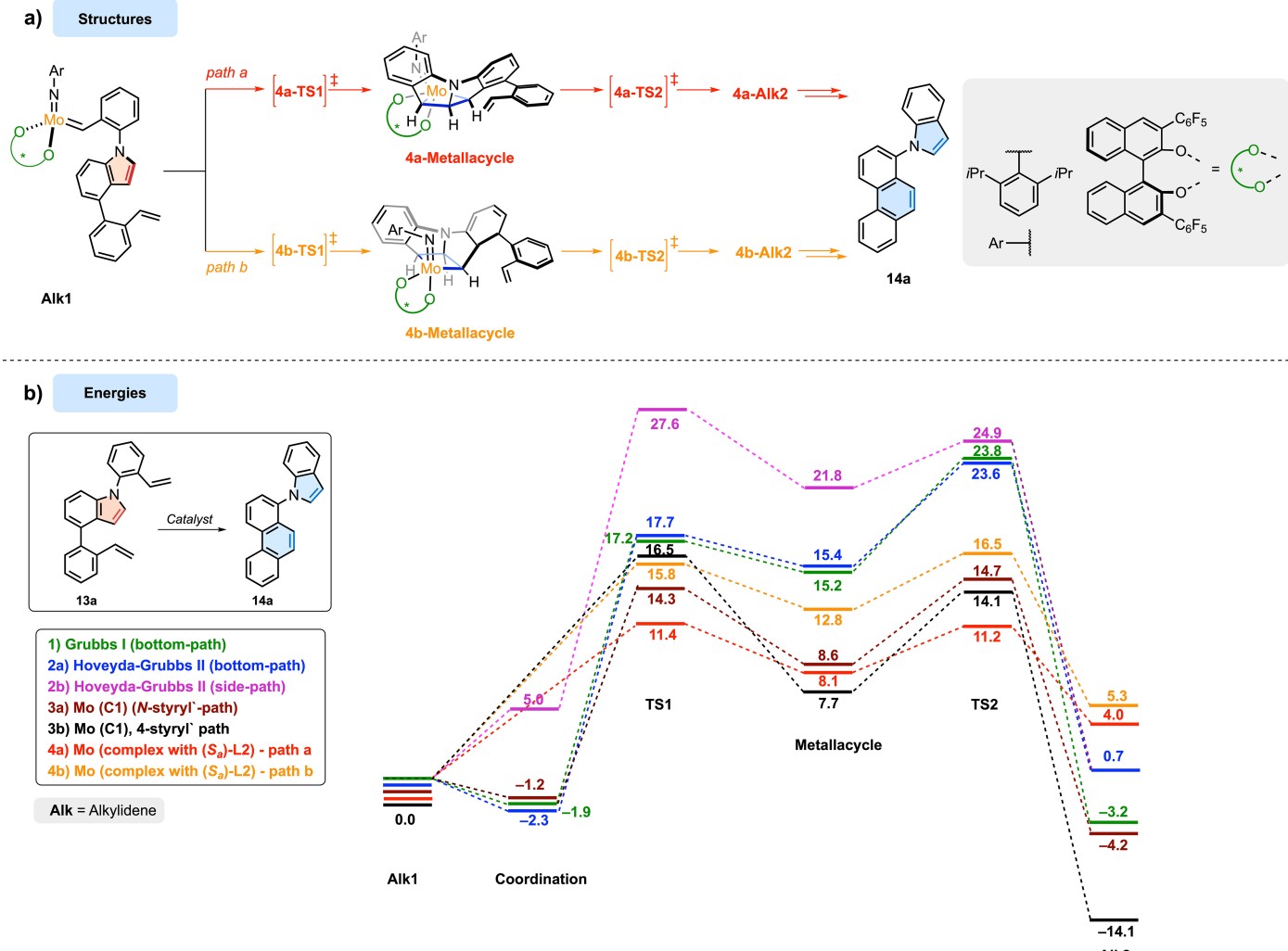

**Extended Data Fig. 3 | Computational studies for ArROM-twofold RCM of substrate 13a for the comparison of different catalytic systems. a**, Catalytic system obtained from Mo-precatalyst and ($S_a$)-**L2**: paths a and b. **b**, Energies in kcal/mol.

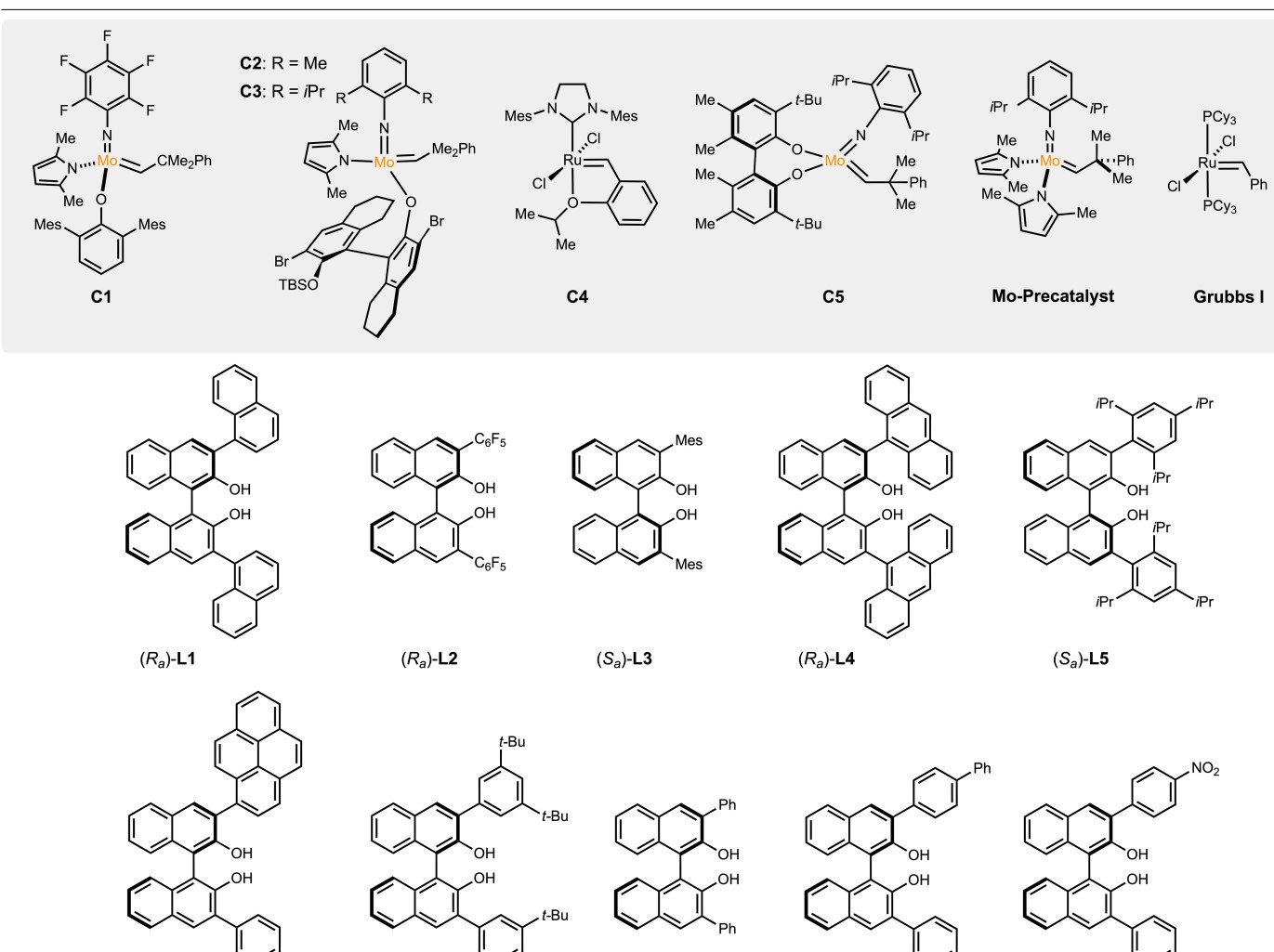

**Extended Data Fig. 4 | Catalysts and ligands utilized for the optimization studies.** Structures in the grey box: catalysts and pre-catalysts. Structures below the grey box: ligands.

**Extended Data Table 1 | Optimization of the reaction conditions for ArROM – twofold RCM for *N*-aryl indoles**

13a 14a

| Entry | Catalyst (mol%) | Ligand (mol%) | Solvent | Temp. [°C] | Conversion[b] [%] | Yield[b] [%] |
|---|---|---|---|---|---|---|
| 1[a] | **C1** (20) | – | Toluene | 65 | >95 | 93 |
| 2[a] | **C2** (10) | – | Toluene | rt | 67 | 47 |
| 3[a] | **C2** (10) | – | Toluene | 65 | >95 | 95 |
| 4[a] | **C2** (10) | – | Toluene | 85 | >95 | 83 |
| 5[a] | **C2** (10) | – | Toluene | 110 | >95 | 48 |
| 6[a] | **C3** (10) | – | Toluene | 65 | 53 | 45 |
| 7[a] | **C4** (10) | – | Toluene | 65 | 63 | 47 |
| 8[a] | **Grubbs I** (10) | – | Toluene | 65 | 15 | 9 |
| 9[a] | **Precat. 1** (10) | ($R_a$)-**L1** | Toluene | 65 | 67 | 67 |
| 10[a] | **Precat. 1** (10) | ($S_a$)-**L2** | Toluene | 65 | >95 | 96 |
| 11[a] | **Precat. 1** (10) | ($R_a$)-**L16** | Toluene | 65 | NR | – |
| 12[c] | **C1** (10) | – | Toluene | 65 | 42 | 42[d] |
| 13[c] | **C1** (15) | – | Toluene | 65 | >95 | 90[d] |
| 14[c] | **C2** (10) | – | Toluene | 65 | >95 | 96[d] |

[a]Reactions were performed on 7.50 μmol scale of **13a** for 18 h. [b]Conversion and yield were determined by ${}^1$H-NMR with durene as an internal standard. [c]Reactions were performed on 70.0 μmol scale. [d]Isolated yield.

**Extended Data Table 2 | Optimization of the reaction conditions for the atroposelective ArROM – RCM cascade**

**15a** ($S_a$) or ($R_a$)-**16a**

| Entry[a] | Catalyst (mol%) | Ligand (mol%) | Solvent | Temp. [°C] | Conversion[b] [%] | Yield[b] [%] | e. r.[c] ($S_a$) : ($R_a$) |
|---|---|---|---|---|---|---|---|
| 1 | **C1** (10) | – | Toluene | 65 | >95 | – | 50 : 50 |
| 2 | **Mo-Precat.** (10) | ($R_a$)-**L1** (30) | Toluene | 65 | >95 | 80 | 6 : 94 |
| 3 | **Mo-Precat.** (10) | ($S_a$)-**L1** (30) | Toluene | 65 | >95 | 76 | 95 : 5 |
| 4 | **Mo-Precat.** (10) | ($R_a$)-**L1** (30) | DCE | 65 | >95 | 70 | 2 : 98 |
| 5 | **Mo-Precat.** (10) | ($R_a$)-**L2** (30) | Toluene | 65 | >95 | 66 | 33 : 67 |
| 6 | **Mo-Precat.** (20) | ($S_a$)-**L3** (60) | Toluene | 65 | 75 | 57 | 97 : 3 |
| 7 | **Mo-Precat.** (30) | ($S_a$)-**L3** (90) | Toluene | 65 | >95 | 54 | 95 : 5 |
| 8 | **Mo-Precat.** (20) | ($R_a$)-**L4** (60) | Toluene | 65 | >95 | 48 | 14 : 86 |
| 9 | **Mo-Precat.** (20) | ($R_a$)-**L4** (60) | Toluene | 50 | >95 | 75 | 13 : 87 |
| 10 | **Mo-Precat.** (30) | ($R_a$)-**L4** (90) | Toluene | 50 | >95 | 60 | 15 : 85 |
| 11 | **Mo-Precat.** (10) | ($S_a$)-**L5** (30) | Toluene | 65 | NR | – | – |
| 12 | **Mo-Precat.** (10) | ($R_a$)-**L6** (30) | Toluene | 65 | >95 | 51 | 4 : 96 |
| 13 | **Mo-Precat.** (10) | ($R_a$)-**L7** (30) | Toluene | 65 | >95 | 67 | 19 : 81 |
| 14 | **Mo-Precat.** (10) | ($R_a$)-**L8** (30) | Toluene | 65 | >95 | 61 | 10 : 90 |
| 15 | **Mo-Precat.** (10) | ($R_a$)-**L9** (30) | Toluene | 65 | >95 | 68 | 15 : 85 |
| 16 | **C5** (20) | – | Toluene | 65 | 4 | 3 | – |

[a]Reactions were performed on 15.0 μmol scale of **15a** for 18 h. [b]Conversion and yield were determined by ¹H-NMR with durene as an internal standard. [c]Determined by HPLC on a chiral stationary phase of the crude product (Chiralpak IC-N3 column, 3 μm, 250x4.6 mm, heptane/iPrOH 97.5:2.5, 1.0 mL/min, 20 °C).