## [Peer Review file · Nature]

Aromatic Ring-Opening Metathesis

Corresponding Author: Professor Christof Sparr

Version 0:

Reviewer comments:

Referee #1

(Remarks to the Author)

This manuscript presents a quite remarkable and highly novel development in alkene metathesis chemistry – specifically, the development of ring-opening / ring-closing ARENE metathesis, a mode of reactivity hitherto thought to be unattainable due to the challenge of breaking aromaticity in the initial carbene metathesis (2+2) step. Under moderate heating, Schrock–Hoveyda catalysts are found to be capable of effecting this sequence, presumably driven by the generation of products that are 'more aromatic' than their starting materials. Several features of this manuscript are noteworthy: a) the synthesis of polycyclic fused aromatics from relatively simple precursors; b) the sequencing of the metathesis cascade to achieve twofold ROM/RCM cascades; c) the equilibration of indoles bearing different substituents; d) the truly remarkable, highly enantioselective ROM/RCM (generation of axial chirality) in the formation of various bi(poly)aryl products.

Given the highly topical nature of controlling axial chirality in materials and medicinal chemistry, and the importance of the Nobel-prize winning metathesis reaction in organic and inorganic chemistry as a whole, one cannot help but feel that this work will be of broad interest to the scientific community, spanning the entirety of the chemical disciplines and beyond. In light of these aspects, and the generality and depth of this study, I am quite certain that this chemistry should be of interest to the readership of Nature.

Some aspects of the manuscript merit further attention:

1. I found the introduction a little unfocused in terms of context and precedent for dearomatisation. It's notable that the present reaction does not, overall, result in dearomatisation — if anything, as noted below, the processes may typically be driven by an increase in aromaticity (or other factors, e.g. prevention of reverse reactions due to strain increase implications). The authors may consider revisiting the introduction in this context, although the fact that dearomatisation is challenging should of course not be ignored.
2. The equilibration of indoles in Figure 2d is fascinating, albeit I did find this part of the Figure a little hard to interpret. However, this reviewer wonders: a) if the reaction would still proceed using stronger EWGs than CF₃ – for example a mesomeric electron withdrawing group (e.g. CO₂Me etc.). In other words, is the CF₃-substituted indole simply unreactive towards the metathesis catalyst, explaining the '0%' yield (or, should this be 100%?); b) Can these equilibria be connected to Hammett substituent constants via a linear free energy relationship? c) In addition, are these product ratios affected by the choice of catalyst?
3. For 11a–k, is any difference in reaction rate noted according to the size of the substituents? The larger the substituent, the more strained / reactive the phenanthrene alkene will become? This reaction is thermoneutral, and so torsional strain in the biaryl (and sm) presumably provide a key driving force.
4. In general, The authors could comment on the driving forces behind specific cascades. Can the (change in) total aromaticity energies be used to explain reactivity, or do the RCM reactions deliver products that are simply kinetically inert under the reaction conditions (as reversibility may involve a large increase in steric strain)? In many cases, the number of aromatic (or partially aromatic) rings in the products increases, which could provide a notable driving force for the reaction, and would allow the practitioner to design new substrates for this chemistry.
5. Somewhere, the authors should comment on the synthesis of the reaction substrates, which appear relatively convenient to prepare by cross-coupling. Some of the synthetic sequences are relatively long and the order of steps appears important, which may be useful to discuss in the Supplementary Material.

6. "the absence of the otherwise notorious fourth ortho- substituent" – to me, the "notorious" phrase does not make best sense.

7. The reaction optimisation is almost entirely confined to the Supplementary Material; it may be of interest to the readership to include some details in the main manuscript. It may also be useful to comment on the scalability of the reaction (I could not easily find a 'larger scale' example, although the scale conducted typically affords >20 mg of products). This is important given some catalyst loadings appear relatively high.

Referee #2

(Remarks to the Author)

This manuscript reports a creative application of molybdenum monoaryloxides pyrrolide catalysts developed extensively by the Hoveyda group in the transformations of one aromatic structure to another. The reaction presumably initiates by forming a Mo-alkylidene on an alkene proximal to an aromatic ring which then results in rupture of the proximal aromatic ring. Coupling the ring-opening process with a ring-closing metathesis to form a more stable aromatic system provides the thermodynamic driving force for the reaction. This reaction is demonstrated on a tetraphene, naphthalene, indoles, benzofuran, and phenanthrenes. The authors further applied the metathesis process to atroposelective biaryl formation using chiral Mo catalysts incorporating axially chiral aryloxide ligands, which have previously been used in other asymmetric ring-opening metathesis reactions. This metathesis approach to biaryl formation provides an alternative to enantioselective cross-coupling reactions that form the chiral axis. Intermolecular (cross-)metathesis between an aromatic ring and a Mo-alkylidene is not demonstrated.

Although the clever substrate design confer interesting transformations overall, the manuscript does not make a compelling case for using this methodology to synthesize the products in Figure 2. The products in Figure 2d-f can be accessed in a more straightforward manner by cross-coupling between a haloarene and an indole. Can the methodology be applied to transform aromatic groups in pharmaceutically relevant compounds? Furthermore, it seems that functional group tolerance would be limited with these highly reactive Mo complexes and the relatively high temperatures employed. If functional group tolerance is low, might there be applications in materials with polyaromatic hydrocarbons benefit from this methodology? It would also be useful to know what aromatic systems are unreactive.

This manuscript does not describe the development of new Mo catalysts to solve reactivity problems– the reactions are enabled by using higher temperatures (65 °C to 120 °C) compared to alkene metathesis (typically rt) on engineered substrates using known Mo catalysts. Mechanistic details about catalyst reactivity are absent in the manuscript. Information about how the catalyst structure affects reactivity towards different aromatic groups is critical for future catalyst development, but this is not present. Computational studies to map out the reaction energetics for metallacyclobutane formation and subsequent collapse for each type of aromatic substrate used should be considered here.

For the reasons stated above, the manuscript falls short of making fundamental breakthroughs on the synthetic and mechanistic fronts. Instead, it is a creative application of Mo-catalyzed metathesis on unconventional substrates and thus may be a better fit for a more specialized journal. The work is well-executed and the Supplementary Information is clearly written with detailed characterization of compounds.

Referee #3

(Remarks to the Author)

This paper is a relatively spectacular demonstration of the power of the olefin metathesis method, but it is worth noting that the authors demonstrate an unusual degree of creativity and a willingness to use air-sensitive catalysts to demonstrate that creativity. In the process they demonstrate that the much-ballyhooed "disadvantages" of Mo catalysts, versus Ru catalysts (although only one side-by-side comparison was made, but I do not necessarily advise any extensive comparison), is nonsense. They have shown that C=C double bonds with a significant degree of aromatic character can indeed take part in a metathesis reaction, if similar bonds are made in the product, thereby making the overall process energetically feasible. Many examples are provided and these first-of-their-kind experiments therefore involve relatively small absolute amounts of catalyst (a few mg), often the relatively standard 5%. There is an enormous amount to do in future studies in terms of substrate and catalyst variations, but they should consider choosing one of the most accessible and potentially scaleable examples in hand now and vary conditions to some degree. For example, is 65° required? Are higher temperatures tolerated? Has any reaction been followed versus time? Can the reactions be scaled up to some degree (usually yields improve and the amount of required catalyst drops)? Etc. All would be informative and would convince the reader that this work is indeed a quantum leap in the application of olefin metathesis to a potentially important area of organic synthesis, at least from my perspective. The heterocycle examples might prove to be especially impressive, so perhaps an example or two of related heterocycles in biology or medicine, or review of same, could be referenced. Overall, my congratulations. This paper certainly is appropriate for publication in Nature. The title is OK, but perhaps it could be spiced up to some degree. It's their call.

Version 1:

Reviewer comments:

Referee #1

(Remarks to the Author)

The authors have responded in full to all questions raised by the reviewers. I particularly appreciated the effort they have taken to expand substrate scope and explore mechanistic aspects, for example through inclusion of the suggested Hammett plot; and also the addition of the extensive computational studies, as well as computations of thermodynamics. These are useful not only in delineating the role of the specific catalysts that can (or cannot) be used, but also in rationalising the driving forces behind this chemistry. As such I feel the manuscript is now well-suited to publication in Nature, subject to a few further minor amendments.

Additional suggested modifications:

- In the introduction, a further fairly standard method to achieve dearomatisation of a benzene ring is through oxidation (e.g. to a quinone).
- " we questioned if an efficient catalytic system could kinetically and thermodynamically address aromatics for ring- opening metathesis if empowered by a suitable endogenous driving force" – it may be useful to add what the 'suitable driving force' is, in this case?
- Related to this point is whether the reader may readily assess the viability of the reaction sequence through a simple comparison of aromatic stabilisation energies between starting material and product (in the absence of ring strain effects as in the phenanthrene examples. This could be a useful predictive tool without the need for extensive computational planning. At the least, Supplementary Table 16 (and indeed the computational work) should be referenced in the main text.

Aside from these minor modifications, the additions made to this manuscript have without doubt improved the depth of this study. I maintain my view that this work will be of wide appeal in opening up a new branch of metathesis chemistry.

Referee #2

(Remarks to the Author)

The revised manuscript by Hutskalova and Sparr makes two key improvements from the original submission: (i) expansion of the reaction scope of the Mo-catalyzed aromatic ring-opening metathesis (ArROM) to include syntheses of polycyclic aromatic hydrocarbons and (ii) DFT computations to predict the outcome of constitutional isomer equilibration and to study the different reaction pathways with various Ru and Mo catalysts. This reviewer agrees with the authors' point that the ability of readily available catalysts to carry out the new transformations reported in the manuscript is powerful. Development of bespoke catalysts may further explore the potential of the ArROM reaction but is best suited for future work. Overall, the revised manuscript demonstrates the utility of the ArROM in three areas of synthesis: (i) skeletal editing of aromatic compounds; (ii) synthesis of challenging polycyclic aromatic compounds; and (iii) atroposelective synthesis of axially chiral compounds. These examples are likely to stimulate readers to apply this work or explore further uses. This reviewer is in favor of publication in Nature.

A few suggested minor changes and/or questions are listed below:

1. The authors should comment on the addition of Figures S8 to S18 in the SI. What question was the authors trying to answer and what conclusions can be obtained from the computations?
2. The authors can consider including the discussion about the driving force for the ArROM in the manuscript and refer the reader to Supplementary Tables 6 and 16.
3. Competition experiment between 15c and 15d (SI page S87): Considering that the two substrates possess different electronic properties on the phenanthrene, can the difference in reactivity be attributed solely to substituent size? A competition experiment between 15a and 15c may be more appropriate.

Basel, 21st October 2024

Dear Editors of Nature,

We are delighted that you and the reviewers find our manuscript of potential interest for publication in Nature. The editorial requests as well as all comments of the reviewers were very helpful and we were able to implement in full. Each of the recommendations improved our manuscript and SI, for which we are most grateful.

Editorial Requests:

The Reviewers all request additional examination of the reaction scope/limitations. Reviewer #2 also asks to look at the application to PAHs, as well as to computationally study how catalyst structure impacts reactivity. Both of these seem worthwhile. Finally, please include reaction optimization as an Extended Data figure, instead of in the SI, as this will form part of the main paper online.

It was possible for us to additionally examine the reaction scope/limitations (**7a-d**→**8a-d**, **9i**→**10i**, **13i**→**14i**) and we looked specifically into the application to PAHs (**7a-d**→**8a-d**, **13i**→**14i**). We are pleased that the developed chemistry can also be successfully implemented for *bidirectional twofold ArROM – twofold RCM* (new Fig. 1d) to a broad diversity of PAH structures: the benzo[*k*]tetraphenes (**8a,b**), benzo[*m*]tetraphene (**8c**), [4]helicenyl-indole (**14i**, Fig. 1g) and even the highly sensitive and unstable dibenzo[*a,j*]tetracene (**8d**). This recommendation was therefore particularly useful as it allowed us to discover the feasibility of ArROM for *bidirectional synthesis* to prepare suitably extended arylated polyaromatics that are not accessible otherwise.

We studied how catalyst structure impacts reactivity by DFT, which compared well with the experimental results. Additionally, energetics consistent with the experimental constitutional isomer equilibration ratios were computationally predicted. Furthermore, strain release as one of the driving forces was examined by DFT studies.

The reaction optimizations (and the computational studies for ArROM) were included as Extended Data Figures.

Referee #1 (Remarks to the Author):

This manuscript presents a quite remarkable and highly novel development in alkene metathesis chemistry – specifically, the development of ring-opening / ring-closing ARENE metathesis, a mode of reactivity hitherto thought to be unattainable due to the challenge of breaking aromaticity in the initial carbene metathesis (2+2) step. Under moderate heating, Schrock–Hoveyda catalysts are found to be capable of effecting this sequence, presumably driven by the generation of products that are 'more aromatic' than their starting materials. Several features of this manuscript are noteworthy: a) the synthesis of polycyclic fused aromatics from relatively simple precursors; b) the sequencing of the metathesis cascade to achieve twofold ROM/RCM cascades; c) the equilibration of indoles bearing different substituents; d) the truly remarkable, highly enantioselective ROM/RCM (generation of axial chirality) in the formation of various bi(poly)aryl products.

Given the highly topical nature of controlling axial chirality in materials and medicinal chemistry, and the importance of the Nobel-prize winning metathesis reaction in organic and inorganic chemistry as a whole, one cannot help but feel that this work will be of broad interest to the scientific community, spanning the entirety of the chemical disciplines and beyond. In light of these aspects, and the generality and depth of this study, I am quite certain that this chemistry should be of interest to the readership of Nature.

Some aspects of the manuscript merit further attention:

1. I found the introduction a little unfocused in terms of context and precedent for dearomatization. It's notable that the present reaction does not, overall, result in dearomatization — if anything, as noted below, the processes may typically be driven by an increase in aromaticity (or other factors, e.g. prevention of reverse reactions due to strain increase implications). The authors may consider revisiting the introduction in this context, although the fact that dearomatization is challenging should of course not be ignored.

We fully agree and are very gratefully for this comment: the focus should much more clearly be kept on *aromatic moieties* rather than the *entire molecule*, since aromatic rings formed at another position of the molecule served as driving force for ArROM. We revised the introduction for dearomatizations accordingly. To emphasize the focus on moieties, this part of the introduction now reads: "Whereas aromatic compounds participate in a multitude of substitution reactions in which the aromaticity of the ring structure is restored, transformations that permanently rupture aromatic moieties, known as dearomatizations, are recognized as highly challenging."

2. The equilibration of indoles in Figure 2d is fascinating, albeit I did find this part of the Figure a little hard to interpret. However, this reviewer wonders: a) if the reaction would still proceed using stronger EWGs than CF₃ – for example a mesomeric electron withdrawing group (e.g. CO₂Me etc.). In other words, is the CF₃-substituted indole simply unreactive towards the metathesis catalyst, explaining the '0%' yield (or, should this be 100%?); b) Can these equilibria be connected to Hammett substituent constants via a linear free energy relationship? c) In addition, are these product ratios affected by the choice of catalyst?

We are very grateful for these constructive and insightful comments on the equilibration of indoles. It is true that Fig. 2d was previously difficult to interpret and we have therefore simplified it accordingly. Additionally, we reorganised the entries with respect to the equilibrium ratios (consistent with question 2b below). Furthermore, equilibria that are distinctly on the side of the substituted indole (**9h** and **9i**) are now described more comprehensively (now Fig. 2e) compared to the previous version of the manuscript.

For 2a) 'the detailed effects of EWGs', 2b) 'the Hammett plots which indeed allow to connect the equilibria to a linear free energy relationship' and 2c) 'the consistent ratios obtained with different catalysts', please see below:

2a) In order to further explore the impact of the EWGs at the indole core, we synthesized the recommended substrate **9i** and subjected it to the constitutional isomer equilibration by means of ArROM using different reaction conditions.

Entry	Catalyst (mol%)	Solvent	Temp. [°C]	9i : 10i
1	C1 (5)	Toluene	65	1 : 0
2	C1 (20)	Toluene	65	1 : 0
3	C2 (20)	Toluene	65	1 : 0

Reactions of **9i** were performed on 7.50 μmol scale for 18 h.

Consistent with the results for the CF₃-bearing indole **9h**, an equilibrium of **9i** ⇌ **10i** on the side of the substituted indole was measured and pure compound **9i** was observed. These experiments confirm that the introduction of EWGs at the indole core shifts the equilibria towards the substituted indoles. To further rationalize the experimental data, we performed DFT calculations to compare the energies between the substituted indoles **9** and the corresponding constitutional isomers **10** (please see the Table below). The calculated Gibbs free energies are in agreement with the experimentally observed positions of the equilibria. The DFT predicted equilibrium ratio for **9h** : **10h** of 1 : 22 for the transformation of the CF₃-substituted indole **9h** is consistent with the experimental

observation when subjecting **9h** to the optimized metathesis conditions. Furthermore, the insignificant energy difference between **9d** and **10d** aligns well with the experimentally observed ratio for **9d** : **10d** of 1 : 1.

Supplementary Table 6:

Density functional theory (DFT) calculations were performed using the Gaussian 16 (revision C.01) program. Geometry optimizations were performed at the B3LYP-D3(BJ) level of theory with the def2-TZVP basis set in the gas phase. Harmonic vibrational frequencies were evaluated for the optimized geometries, with minima characterized by the absence of imaginary frequencies. Quasiharmonic corrections to the entropy for frequencies below 100 cm⁻¹ were calculated with the Goodvibes program by employing the method of Grimme. Single point energies were calculated at B3LYP-D3(BJ) level of theory with the def2-TZVP basis set with implicit solvation model SMD for toluene. The optimized structures were visualized in CYLView.

2b) We furthermore thank the reviewer for the thoughtful suggestion to connect the equilibria to Hammett substituents coefficients. Indeed, a linear correlation is observed when the logarithm of the experimentally observed equilibrium constants is plotted against the substituent constant σ . The negative slope indicates that the presence of EWGs disfavours the equilibration of **9** towards the corresponding isomer **10** which is also consistent with the experimental data for CF₃- and CO₂Me-substituted indole derivatives **9h** and **9i**. Moreover, a linear correlation was also obtained between DFT calculated Gibbs free energies of the equilibria and the Hammett substituent coefficients.

Supplementary Table 7:

Hammett plot for experimental equilibrium constants

Hammett plot for DFT calculated Gibbs free energies

2c) To address whether the product ratios might be influenced by the choice of catalyst, we performed experiments using different catalysts (**C1** and **C2**), varying their loading across different substrates. The results of our previous investigations conducted for the indole **9d** show that neither change of the catalyst, nor alteration of the catalyst loading (5 / 20 mol%) or temperature (65 / 85°C) had an impact on the ratio. The same observation was made for substrates **9f** and **9b**, confirming that the ratios remained unaffected by the different catalysts and their loading.

Supplementary Table 5:

Entry	Catalyst (mol%)	Solvent	Temp. [°C]	(9f/b : 10f/b) ^c
1 R=Cl ^a	C1 (20)	Toluene	65	1 : 0.4
2 R=Cl ^a	C2 (5)	Toluene	65	1 : 0.4
3 R=Cl ^a	C2 (20)	Toluene	65	1 : 0.4
4 R=Me ^b	C1 (20)	Toluene	65	1 : 2.4
5 R=Me ^b	C2 (5)	Toluene	65	1 : 2.4
6 R=Me ^b	C2 (20)	Toluene	65	1 : 2.3

^aReactions were performed on 7.50 μmol scale of **9f** for 18 h. ^bReactions were performed on 7.50 μmol scale of **9b** for 18 h. ^cRatios were determined by ¹H-NMR.

3. For 11a–k, is any difference in reaction rate noted according to the size of the substituents? The larger the substituent, the more strained / reactive the phenanthrene alkene will become? This reaction is thermoneutral, and so torsional strain in the biaryl (and sm) presumably provide a key driving force.

Note: compounds **11a–k** are now **15a–k** in the revised version of the manuscript.

To investigate this interesting kinetic aspect, we performed an additional competition experiment, where a 1:1 mixture of phenanthrene substrates **15c** and **15d** was subjected to ArROM under the optimized conditions for 15 minutes (please see the Scheme below). ¹H NMR analysis of the crude reaction mixture revealed that full conversion was achieved for the transformation of **15d** to **16d**, while only 80% conversion was observed for the substrate **15c**. The obtained data reflect the differences in reaction rates of **15c** and **15d** which correlate with the size of the substituents. Indeed, the larger size of the methoxy-group in **15d** compared to a hydrogen atom in **15c** is expected to create more strain: as a result, an increased reaction rate was observed. Further findings relevant for the driving force of the reactions were obtained while answering the next question.

In the SI after Supplementary Table 12:

Competition experiment (role of strain-release)

Reaction conditions of the competition experiment: **11c** (7.50 μmol), **11d** (7.50 μmol), Mo-Precatalyst (0.750 μmol), (R_a) -L1 (2.25 μmol), toluene (0.5 mL), 65°C.

4. In general, The authors could comment on the driving forces behind specific cascades. Can the (change in) total aromaticity energies be used to explain reactivity, or do the RCM reactions deliver products that are simply kinetically inert under the reaction conditions (as reversibility may involve a large increase in steric strain)? In many cases, the number of aromatic (or partially aromatic) rings in the products increases, which could provide a notable driving force for the reaction, and would allow the practitioner to design new substrates for this chemistry.

We thank the reviewer for raising this important question. Not only the number of aromatic rings, but also the type of aromatic system, degree of aromatic stabilization and the strain release allow to identify suitable substrates for this chemistry. To further explore the driving force of the reactions, we set out to perform DFT calculations to compare Gibbs free energies of selected pairs of different types of starting materials and their corresponding products. N.B. negative values for the Gibbs free energy differences indicate a thermodynamically favourable (exothermic) formation of the products in contrast to the reverse reaction, which is particularly relevant for the cases where the product contains a reactive vinyl group.

Supplementary Table 16:

Density functional theory (DFT) calculations were performed using the Gaussian 16 (revision C.01) program. Geometry optimizations were performed at the B3LYP-D3(BJ) level of theory with the def2-TZVP basis set in the gas phase. Harmonic vibrational frequencies were evaluated for the optimized geometries, with minima characterized by the absence of imaginary frequencies. Quasiharmonic corrections to the entropy for frequencies below 100 cm⁻¹ were calculated with the Goodvibes program by employing the method of Grimme. Single point energies were calculated at B3LYP-D3(BJ) level of theory with the def2-TZVP basis set with implicit solvation model SMD for toluene. The optimized structures were visualized in CYLView.

5. Somewhere, the authors should comment on the synthesis of the reaction substrates, which appear relatively convenient to prepare by cross-coupling. Some of the synthetic sequences are relatively long and the order of steps appears important, which may be useful to discuss in the Supplementary Material.

We are grateful for this helpful advice and added the following comment for the substrate synthesis in the revised version: "...substrates **15a-k**, which are readily accessible by cross-coupling chemistry." for AAROM. The synthetic sequences with the order of steps to the precursors were summarized for the different types of substrates in the Supplementary Material (e.g. to prepare the additional substrates for the PAHs **7a**, **7b**, **7c** or **7d**).

6. "the absence of the otherwise notorious fourth ortho-substituent" – to me, the "notorious" phrase does not make best sense.

We fully agree and revised this phrase accordingly to: "Interestingly, a decrease of selectivity was observed for atropisomer (*R*_a)-**16c** devoid of an otherwise cumbersome fourth biaryl ortho-substituent."

7. The reaction optimisation is almost entirely confined to the Supplementary Material; it may be of interest to the readership to include some details in the main manuscript. It may also be useful to comment on the scalability of the reaction (I could not easily find a 'larger scale' example, although the scale conducted typically affords >20 mg of products). This is important given some catalyst loadings appear relatively high.

We are grateful for this suggestion to better highlight the reaction optimization: it is now included as Extended Data Figure 1. We also appreciate bringing up the importance of scalability. The majority of our substrate scope exploration was performed on 70.0 μmol scale, which generally affords the products in the range of 15-35 mg. To further showcase the practicality of the developed methodologies, we performed two ArROM reactions on larger scale: the ArROM-twofold RCM of *N*-aryl indole **13a** and the atroposelective ArROM-twofold RCM cascade of **17a**. To our delight, both of the transformations proved to be highly efficient on > 110 mg scale, yielding the desired products with excellent yields and remarkably high atroposelectivity (> 99:1 e.r.) for **18a**. It is also worth pointing out that the reactions on larger scale (as predicted by referee #3) were successfully performed with decreased catalyst loadings: 2.0 mol% and 5.0 mol% for **13a** and **17a**, respectively (please see the Scheme below). The new results for the scale-up experiments are described in the SI and are both highlighted in the manuscript (text, Fig. 2g and Fig. 4a).

Referee #2 (Remarks to the Author):

This manuscript reports a creative application of molybdenum monoaryloxides pyrrolide catalysts developed extensively by the Hoveyda group in the transformations of one aromatic structure to another. The reaction presumably initiates by forming a Mo-alkylidene on an alkene proximal to an aromatic ring which then results in rupture of the proximal aromatic ring. Coupling the ring-opening process with a ring-closing metathesis to form a more stable aromatic system provides the thermodynamic driving force for the reaction. This reaction is demonstrated on a tetraphene, naphthalene, indoles, benzofuran, and phenanthrenes. The authors further applied the metathesis process to atroposelective biaryl formation using chiral Mo catalysts incorporating axially chiral aryloxide ligands, which have previously been used in other asymmetric ring-opening metathesis reactions. This metathesis approach to biaryl formation provides an alternative to enantioselective cross-coupling reactions that form the chiral axis. Intermolecular (cross-)metathesis between an aromatic ring and a Mo-alkylidene is not demonstrated.

Although the clever substrate design confer interesting transformations overall, the manuscript does not make a compelling case for using this methodology to synthesize the products in Figure 2. The products in Figure 2d-f can be accessed in a more straightforward manner by cross-coupling between a haloarene and an indole. Can the methodology be applied to transform aromatic groups in pharmaceutically relevant compounds? Furthermore, it seems that functional group tolerance would be limited with these highly reactive Mo complexes and the relatively high temperatures employed. If functional group tolerance is low, might there be applications in materials with polyaromatic hydrocarbons benefit from this methodology? It would also be useful to know what aromatic systems are unreactive.

We are grateful to the reviewer for the valuable feedback and advice. We have now additionally examined the scope/limitations and looked into the application to PAHs that are not accessible otherwise. Besides the many other products prepared by ArROM that are not readily accessible by cross-coupling chemistry, the main objective in Fig. 2d-f (now Fig. 2e-g) is to demonstrate that ArROM reliably cleaves (and even equilibrates) most important aromatic ring structures in a universal manner, including pertinent heterocyclic compounds that are very frequently incorporated as core units of common pharmaceuticals or ingredients for various other applications.

For the functional group tolerance, we observed that Mo-catalysed ArROM is compatible with a diversity of functionalities, including ethers, esters, Cl, F, heterocycles, acetals, CF₃ and even Br or sensitive PAHs. For future applications to products with low thermal stability, we performed additional experiments at reduced temperatures. Indeed, we were pleased to observe that **19c** reacted by an atroposelective ArROM to form (*R_a*)-**20c** even at room temperature with 75% conversion. Similarly, **14a** was obtained from the corresponding alkene precursor **13a** at room temperature instead of 65°C with 67% conversion (please also see the response to referee #3). These results confirm that particularly mild conditions for (atroposelective) ArROM are feasible for thermally highly sensitive substrates if the need arises. The new experiments were added to the revised version of the Supplementary Table 9 (**14a**) and Supplementary Table 15 ((*R_a*)-**20c**).

We also thank the reviewer for raising the interesting question about the implementation of ArROM in the field of polyaromatic hydrocarbons. Together with the previous examples for polyaromatic hydrocarbons (e.g. PAHs **2** or **4**), we now further illustrate the versatility of the approach by applying ArROM to form extended PAHs, such as benzo[*k*]tetraphene derivatives **8a** and **8b**. To our delight, both alkene precursors **7a** and **7b**, which were synthesized from readily accessible starting materials, were converted by another strategy for aromatic ring opening metathesis, the *bidirectional twofold ArROM – twofold RCM*, in high yields leading to distinct PAHs. Notably, also the arylated benzo[*m*]tetraphene and even the unstable dibenzo[*a,j*]tetracene **8d**, for which we observed degradation upon isolation in a pure NMR sample over hours, were accessible. While the method is capable of providing access to such unique structures with low stability, the limitation in yield can be attributed to the spontaneous decomposition of the PAH **8d** by itself. Additionally, we investigated the reaction of another new indole substrate **13i** which was subjected to ArROM giving [4]helicenyl-indole **14i** with an exceptionally high isolated yield (98%). We are thus optimistic that these examples and their diversity suitably demonstrate that ArROM is likely to become a most valuable tool to access unique PAHs that are otherwise challenging to synthesize.

Unreactive substrates, respectively substrates which possess or lack a suitable driving force, as also pointed out by referee #1 in question 4, are delineated by considering the number of aromatic rings, the type of the aromatic system, degree of aromatic stabilization and the strain release, as now predicted by the DFT calculations and discussed above for question 4 of referee #1 together with further computational results of the processes below.

This manuscript does not describe the development of new Mo catalysts to solve reactivity problems— the reactions are enabled by using higher temperatures (65 °C to 120 °C) compared to alkene metathesis (typically rt) on engineered substrates using known Mo catalysts. Mechanistic details about catalyst reactivity are absent in the manuscript. Information about how the catalyst structure affects reactivity towards different aromatic groups is critical for future catalyst development, but this is not present. Computational studies to map out the reaction energetics for metallacyclobutane formation and subsequent collapse for each type of aromatic substrate used should be considered here.

We thank the reviewer for this comment. Having observed that ArROM can take place at low temperature, we consider the remarkable performance of the readily available and commercial catalysts to be a great advantage, as it ensures an immediate adoption of aromatic ring-opening metathesis (ArROM) by a broad scientific community. It is another testimony of the monumental developments for these catalysts to reach such broad generality, even for newly emerging methods.

To computationally study how catalyst structure impacts reactivity, we performed DFT calculations to investigate the energetics of the reaction profiles for different Mo- and Ru-catalysts (such as Grubbs I, Hoveyda-Grubbs II, **C1**, and the catalysts in situ generated with precatalyst and (*S_o*)-**L2**) with **13a** as a substrate (please see the Schemes below). In particular, the higher Gibbs energy of activation to TS1 for Ru catalysts (Grubbs I, Hoveyda-Grubbs II) compared to Mo-catalysts are consistent with the higher reactivity with the latter. For example, while **C1** was forming the desired product **14a** with full conversion in excellent yield, only 9% and 47% yield were obtained for Grubbs I and Hoveyda-Grubbs II catalysts, respectively. To provide more insight into the mechanism of the ArROM for Ru catalysts, we also compared the side- and bottom-path routes for the Hoveyda-Grubbs II catalyst. The obtained reaction energetics indicate a preference for the bottom-path reaction, which is reflected by a higher energy of TS1 for the side-path mechanism. Additionally, since two vinyl groups are present in **13a** and both could initiate the reaction, we computed both potential directions of the reaction; specifically, which vinyl group initiates metathesis. To achieve this, we compared two alternative routes for molybdenum catalyst **C1**: the *N*-styryl'- and 4-styryl' paths. The comparison of the two resulting reaction profiles indicates that both of them are feasible, while the reaction starting at the *N*-styryl' group has a lower Gibbs energy of activation for TS1.

Comparison of Different Metathesis Catalysts:

ArROM-twofold RCM for **13a** with different catalytic systems: a) Ru-catalyzed metathesis (Grubbs I and Hoveyda-Grubbs II catalysts): bottom- and side-path mechanisms; b) Mo-catalyzed metathesis (C1 catalyst): *N*-styryl'- and 4-styryl'-paths. c) Catalytic system obtained from Mo-Precatalyst and (S_O)-L2: paths a and b.

Density functional theory (DFT) calculations were performed using the Gaussian 16 (revision C.01) program. Geometries of intermediates and transition states were optimized using the dispersion-corrected B3LYP-D3 functional. The SDD basis set was used for Mo/Ru atoms, while the 6-31G(d) basis set was applied for other atoms. Vibrational frequency calculations were performed at the same level of theory of the optimization to confirm if each structure is a local minimum (zero imaginary frequencies) or a transition state (one imaginary frequency). Single-point energy calculations were carried out using the M06 functional with the SDD basis set for Mo/Ru atoms and 6-311+G(d,p) basis set for other atoms. Solvent effects were accounted for with the SMD model of toluene, which was the solvent of choice in experiments. Quasi-harmonic corrections to the entropy for frequencies below 100 cm⁻¹ were calculated with the Goodvibes program by employing the method of Grimme.

DFT-calculated Gibbs free energies (kcal/mol) of the reaction profiles of **13a** with different Mo- and Ru-based catalysts. SMD(toluene)/M06/SDD(Mo/Ru)-6-311+G(d,p)//B3LYP-D3(BJ)/SDD(Mo/Ru)-6-31G(d)

For the reasons stated above, the manuscript falls short of making fundamental breakthroughs on the synthetic and mechanistic fronts. Instead, it is a creative application of Mo-catalyzed metathesis on unconventional substrates and thus may be a better fit for a more specialized journal. The work is well-executed and the Supplementary Information is clearly written with detailed characterization of compounds.

We hope these revisions, for both aspects – the synthetic and mechanistic–, the preparation of the otherwise inaccessible PAHs and the DFT study on how catalyst structure impacts reactivity, fully resolves these concerns. We highly appreciate the reviewer's comment that the work is well executed and that the Supplementary Information is clearly written with detailed characterisation.

Referee #3 (Remarks to the Author):

This paper is a relatively spectacular demonstration of the power of the olefin metathesis method, but it is worth noting that the authors demonstrate an unusual degree of creativity and a willingness to use air-sensitive catalysts to demonstrate that creativity. In the process they demonstrate that the much-ballyhooed "disadvantages" of Mo catalysts, versus Ru catalysts (although only one side-by-side comparison was made, but I do not necessarily advise any extensive comparison), is nonsense. They have shown that C=C double bonds with a significant degree of aromatic character can indeed take part in a metathesis reaction, if similar bonds are made in the product, thereby making the overall process energetically feasible. Many examples are provided and these first-of-their-kind experiments therefore involve relatively small absolute amounts of catalyst (a few mg), often the relatively standard 5%. There is an enormous amount to do in future studies in terms of substrate and catalyst variations, but they should consider choosing one of the most accessible and potentially scaleable examples in hand now and vary conditions to some degree. For example, is 65° required? Are higher temperatures tolerated? Has any reaction been followed versus time? Can the reactions be scaled up to some degree (usually yields improve and the amount of required catalyst drops)? Etc. All would be informative and would convince the reader that this work is indeed a quantum leap in the application of olefin metathesis to a potentially important area of organic synthesis, at least from my perspective.

We highly appreciate the enthusiastic and thoughtful comments of Referee #3. We are very grateful for the recommendation to further investigate one of the ArROM examples in terms of scalability and variation of conditions. To implement these suggestions, we further probed the reaction conditions for the ArROM-twofold RCM of *N*-aryl indole **13a**. Besides testing the possibility for a particularly mild reaction (entry 11), the tolerance for high temperatures was investigated by performing the reaction at 85°C and 110°C (entries 9, 10).

Entry	Catalyst (mol%)	Ligand	Solvent	Temp. [°C]	Conversion ^b [%]	Yield ^b [%]
1 ^a	C1 (10)		Toluene	65	>95	76
2 ^a	C1 (20)		Toluene	65	>95	93
3 ^a	C4 (10)		Toluene	65	63	47
4 ^a	C3 (10)		Toluene	65	53	45
5 ^a	C2 (10)		Toluene	65	>95	95
6 ^c	C1 (10)		Toluene	65	42	42 ^d
7 ^c	C1 (15)		Toluene	65	>95	90 ^d
8 ^c	C2 (10)		Toluene	65	>95	96 ^d
9 ^a	C2 (10)		Toluene	85	>95	83
10 ^a	C2 (10)		Toluene	110	>95	48
11 ^a	C2 (10)		Toluene	rt	67	47
12 ^a	Grubbs I		Toluene	65	15	9
13 ^a	Precat. 1(10)	(R_a)-L1	Toluene	65	67	67
14 ^a	Precat. 1(10)	(S_a)-L2	Toluene	65	>95	96
15 ^a	Precat. 1(10)	(R_a)-L13	Toluene	65	NR	—
					(90% recovered)	

^aReactions were performed on 7.50 μmol scale of **9a** for 18 h. ^bConversion and yield were determined by ¹H-NMR with durene as an internal standard. ^cReactions were performed on 70.0 μmol scale. ^dIsolated yield.

While full conversion and a good yield (83%) were achieved at 85°C, a moderate yield of 48% was obtained at 110°C, which could be explained by a potential decomposition of styrene **13a**. We were pleased to observe that **13a** undergoes ArROM to **14a** at room temperature with 67% conversion and 47% yield (18 h, entry 11). Additionally, we extended our investigation of this transformation by assessing the reactivity with other catalysts. For example, while only 9% of the product was obtained with Grubbs I catalyst, Mo-complexes obtained from the precatalyst and BINOLs ((*R_a*)-L1 and (*S_a*)-L2) showed significantly better performance with good (67%) and excellent (96%) yields for **14a**, respectively. The observed difference in reactivity of Ru- and Mo-based catalysts stands

in line with the reaction energetics that were obtained by DFT calculations. Indeed, the significantly lower energies of the TS1 and the metallacycle intermediate for Mo catalysts compared to Ru ones, indicate a higher reactivity with the Mo-complexes. To confirm the practical utility of the developed methodologies, we conducted scale-up experiments with *N*-aryl indole **13a** (please also see the answer to question 7 of referee #1). As predicted by referee #3, the reaction was successfully performed with an excellent yield even with a reduced catalyst loading (2.0 mol%, 97%). The scalability of atroposelective ArROM was also explored with substrate **17a**. A remarkably high yield and stereoselectivity were achieved with 5 mol% of the catalyst (97%, >99 : 1 e.r.). Next, we set out to follow ArROM reaction of **13a** over time. The obtained data illustrates that 92% conversion is achieved after 9 hours and that the reaction is complete within a total of 18 hours.

In the SI after General Procedure K / product 14a: Reaction monitoring (0.35 mmol scale, 2 mol% catalyst C2)

Reaction time min	Conversion %
0	0
10	2
70	16
100	24
120	30
165	40
250	63
295	69
320	74
340	76
380	80
435	85
500	89
540 (9 hours)	92
1080 (18 hours)	100

The heterocycle examples might prove to be especially impressive, so perhaps an example or two of related heterocycles in biology or medicine, or review of same, could be referenced.

We highly appreciate this comment and now cite the excellent book by G. W. Gribble, *Indole Ring Synthesis: From Natural Products to Drug Discovery*. John Wiley & Sons, Chichester, West Sussex, UK (2016), reference 44.

Overall, my congratulations. This paper certainly is appropriate for publication in Nature. The title is OK, but perhaps it could be spiced up to some degree. It's their call.

We are very grateful for the recommendation for publication in Nature and the compliments. We considered several other titles, but *aromatic ring opening metathesis* appeared more suitable since it combines *arene and heteroarene metathesis*, thereby referring to the ring opening of carbo- and heterocyclic aromatics.

Overall, we highly appreciate each and every of the comments which encouraged these improvements. We are pleased that the revisions outlined above allowed us to answer the editorial requests as well as the comments of the reviewers in full and are grateful that you are further considering our revised manuscript for publication in Nature.

Yours sincerely,
 Valeriia Hutskalova
 Christof Sparr

Basel, 25th November 2024

Dear Editors of Nature,

We are delighted that you and the reviewers find our revised version of the manuscript in principle suitable for publication in Nature. In this second revised version, we addressed all remaining points of the reviewers.

Referee #1 (Remarks to the Author):

The authors have responded in full to all questions raised by the reviewers. I particularly appreciated the effort they have taken to expand substrate scope and explore mechanistic aspects, for example through inclusion of the suggested Hammett plot; and also the addition of the extensive computational studies, as well as computations of thermodynamics. These are useful not only in delineating the role of the specific catalysts that can (or cannot) be used, but also in rationalising the driving forces behind this chemistry. As such I feel the manuscript is now well-suited to publication in Nature, subject to a few further minor amendments.

Additional suggested modifications:

- In the introduction, a further fairly standard method to achieve dearomatisation of a benzene ring is through oxidation (e.g. to a quinone).*

We thank referee #1 for this very valuable recommendation. We correspondingly modified the introduction by mentioning oxidative dearomatizations: "Important dearomatization methods comprise the Birch reduction¹², arene hydrogenations^{13–16}, a diversity of cycloadditions^{17,18} and oxidations¹⁹, such as the formation of quinones.

- " we questioned if an efficient catalytic system could kinetically and thermodynamically address aromatics for ring- opening metathesis if empowered by a suitable endogenous driving force" – it may be useful to add what the 'suitable driving force' is, in this case?*

We agree with the referee #1 and specified the studied driving forces in the following way:

"In particular, we envisaged that the formation of stabilized aromatic ring systems and strain release would enable arene and heteroarene cleavage."

- Related to this point is whether the reader may readily assess the viability of the reaction sequence through a simple comparison of aromatic stabilisation energies between starting material and product (in the absence of ring strain effects as in the phenanthrene examples. This could be a useful predictive tool without the need for extensive computational planning. At the least, Supplementary Table 16 (and indeed the computational work) should be referenced in the main text.*

We highly appreciate this suggestion from the referee #1. The following comments on the computational work were added to the main text:

"The thermodynamically favourable formation of product 2 associates with its higher overall aromatic stabilization compared to substrate 1 (for DFT calculations, see Supplementary Table 16)."

Aside from these minor modifications, the additions made to this manuscript have without doubt improved the depth of this study. I maintain my view that this work will be of wide appeal in opening up a new branch of metathesis chemistry.

Referee #2 (Remarks to the Author):

The revised manuscript by Hutskalova and Sparr makes two key improvements from the original submission: (i) expansion of the reaction scope of the Mo-catalyzed aromatic ring-opening metathesis (ArROM) to include syntheses of polycyclic aromatic hydrocarbons and (ii) DFT computations to predict the outcome of constitutional isomer equilibration and to study the different reaction pathways with various Ru and Mo catalysts. This reviewer agrees with the authors' point that the ability of readily available catalysts to carry out the new transformations reported in the manuscript is powerful. Development of bespoke catalysts may further explore the potential of the ArROM reaction but is best suited for future work. Overall, the revised manuscript demonstrates the utility of

the ArROM in three areas of synthesis: (i) skeletal editing of aromatic compounds; (ii) synthesis of challenging polycyclic aromatic compounds; and (iii) atroposelective synthesis of axially chiral compounds. These examples are likely to stimulate readers to apply this work or explore further uses. This reviewer is in favor of publication in Nature.

A few suggested minor changes and/or questions are listed below:

1. The authors should comment on the addition of Figures S8 to S18 in the SI. What question was the authors trying to answer and what conclusions can be obtained from the computations?

We are grateful to the referee #2 for raising this question. The Non-Covalent Interactions (NCI) analysis of the metallacyclobutanes was performed to determine NCIs that could contribute to the stabilization of these intermediates. These visualizations might be valuable for catalyst design in the future. To highlight the indicated NCIs and differences among the analysed metallacyclobutanes, we added the following note to the corresponding section of the Supplementary Information:

“Note: NCI analysis of a selection of metallacycles (**2a/b**, **3a/b**, **4a/b**) was performed to determine the major NCIs that could contribute to the energy differences between these reaction intermediates. The commonly observed NCIs included C–H \cdots π and π – π interactions. For example, stronger π – π interactions observed for the metallacycle **3b** (Figure S11) in comparison to the metallacycle **3a** (Figure S10) could partially explain its higher stability. On the other hand, metallacycle-**4a** possesses more C–H \cdots π interactions (Figure S11) than metallacycle **4b** (Figure S12), which might be among the factors causing its lower Gibbs free energy.”

2. The authors can consider including the discussion about the driving force for the ArROM in the manuscript and refer the reader to Supplementary Tables 6 and 16.

We also highly appreciate this suggestion of referee #2, which we implemented as described above in the response to the recommendation 3 of referee #1.

3. Competition experiment between 15c and 15d (SI page S87): Considering that the two substrates possess different electronic properties on the phenanthrene, can the difference in reactivity be attributed solely to substituent size? A competition experiment between 15a and 15c may be more appropriate.

We agree with the referee #2 that electronic properties of the substituent could also affect the difference in reactivity together with the strain release, which provides an important driving force. To address this question, we additionally performed DFT computations to estimate the relative energy of **15d** and **16d**. Interestingly, the calculations suggest that the electronic effect has a lower impact and that both **15a/16a** or **15d/16d** have both a significantly higher energy difference compared to **15c/16c** (Supplementary table 16 contains these additional computations).

We highly appreciate the reviewers' and editor's insightful suggestions. We are delighted and grateful that our manuscript is accepted in principle for publication in Nature.

Yours sincerely,
Valeriia Hutskalova
Christof Sparr